# Vision-R1: Evolving Human-Free Alignment in Large Vision-Language Models via Vision-Guided Reinforcement Learning

## Abstract

Large Vision-Language Models (LVLMs) typically follow a two-stage training paradigm—pretraining and supervised fine-tuning. Recently, preference optimization, derived from the language domain, has emerged as an effective post-training reinforcement strategy to enhance the capabilities of LVLMs. However, constructing high-quality human-annotated preference data and developing robust reward models to mimic these preferences are both costly and challenging. Motivated by this observation, we propose Vision-R1, a novel vision-guided R1-like reinforcement learning algorithm for LVLMs that rewards models with definitive vision feedback. It only leverages curated instruction data, eliminating the need for specialized reward models and handcrafted preference datasets. We incorporate a criterion-driven reward function that further integrates multi-dimensional feedback to evaluate model completions comprehensively based on the vision task logic. Furthermore, we introduce a progressive rule refinement strategy that dynamically adjusts the reward criteria during training, enabling continuous model improvement and mitigating reward hacking. Extensive experiments on both in-distribution and out-of-distribution benchmarks demonstrate that fine-tuning the 7B LVLMs with Vision-R1 achieves consistent performance gains, with even up to 50% gains.

## 1 Introduction

Recently, notable progress has been made in Large Vision Language Models (LVLMs) (Chen et al., 2024b; Liu et al., 2023; Li et al., 2023a; Liu et al., 2024b; Bai et al., 2023; Tong et al., 2024), which encode images into textual tokens and respond to instructions based on visual cues. These models typically follow a two-stage training paradigm, where pretraining establishes a foundational understanding of visual information, while supervised fine-tuning (Liu et al., 2023) enhances their ability to follow instructions and solve problems. Through this process, advanced LVLMs have shown remarkable potential in integrating vision and language to address complex tasks.

Despite these advancements, LVLMs still fall short of meeting human expectations as effectively as Large Language Models (LLMs) (Achiam et al., 2023; Brown et al., 2020; Touvron et al., 2023; Liu et al., 2024a), primarily due to limitations in vision-language data. To bridge this gap, preference optimization(Sun et al., 2023; Xiong et al., 2024; Dong et al., 2024; Zhang et al., 2025), derived from LLMs(Ouyang et al., 2022; Chen et al., 2024c; Rafailov et al., 2023) for its data efficiency and performance benefits, has been introduced as a post-training reinforcement strategy to refine LVLM responses based on human feedback. Although these methods reduce data consumption to the thousand-level, constructing high-quality vision-language preference datasets remains resource-intensive. Meanwhile, training a reliable reward model to capture nuanced preferences with varying subjectivity remains a major challenge.

With the success of LLM Deekseek-R1 (Guo et al., 2025), the rule-based Group Relative Policy Optimization (GRPO) (Shao et al., 2024) algorithm offers a new approach to track this challenge. While previously validated in reasoning tasks such as math (Shao et al., 2024) and code (Guo et al., 2024), R1 model further prove that rule-based rewards enhance comprehension and reasoning across multiple domains, enabling both reasoning and non-reasoning tasks performance improvement. Moreover, with the incorporation of visual information, vision-language question-answer data becomes more

objective and definitive, providing clearer solutions and cues. Existing human-annotated instruction data (Li et al., 2024; Zhan et al., 2024b) naturally provide precise responses that align with human preferences. This raises a critical question: *Can an R1-like reinforcement learning method further enhance LVLM capabilities with curated vision-language instruction data?*

In this paper, we propose Vision-R1, a novel vision-guided R1-like reinforcement learning algorithm for LVLMs that eliminates the need for specialized reward models and handcrafted preference datasets. To achieve this, we conduct a comprehensive investigation into reward modeling and training strategy, as indicated in Figure 1. We first introduce a **criterion-driven reward function** to quantitatively evaluate each completion based on visual feedback, providing an objective standard for absolute rewards without relatively ranking based on preference data. This function delivers multi-dimensional reward signals guided by vision task criteria, such as

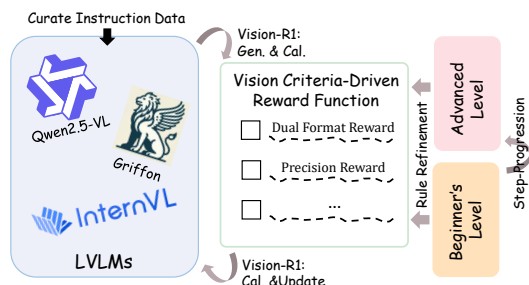

Figure 1: Key designs of Vision-R1.

precision to measure box accuracy via transforming textually numerical tokens to coordinates. Our design enables the model to develop a deeper understanding of task characteristics and generate more accurate responses, surpassing the token-level supervision used in SFT that ignores spatial identity. Building on the reward modeling, we further introduce a **progressive rule refinement strategy** that dynamically adjusts reward criteria throughout training to facilitate continuous improvement. Inspired by curriculum learning (Bengio et al., 2009) and human learning processes, this strategy follows two key principles: differentiation and staged progression. This differentiation mechanism encourages the model to continuously refine its predictions for optimal performance. Meanwhile, training is structured into beginner and advanced phases with progressively stricter reward criteria in the advanced phase to prevent reward hacking and ensure sustained progression.

To validate the effectiveness of our approach, we train two advanced LVLMs, Griffon-G-7B (Zhan et al., 2024a) and Qwen2.5-VL-7B (Bai et al., 2025), on the curated data and evaluate them across multiple in-domain and out-of-domain object localization tasks, as well as general QA benchmarks. Extensive experiments demonstrate that: (1) Vision-R1 achieves **significant performance enhancement** across diverse tasks, including wild visual grounding and dense object detection, with even up to 50% improvement for Qwen2.5-VL. (2) Compared to SFT, Vision-R1 demonstrates **better generalization capabilities** with an average of 6% improvement on unseen scenarios while maintaining advanced QA capabilities.

## 2 RELATED WORKS

### 2.1 LARGE VISION LANGUAGE MODELS

In recent years, LVLMs(Chen et al., 2024b; Liu et al., 2023; Li et al., 2023a; Liu et al., 2024b; Tong et al., 2024; Steiner et al., 2024; Bai et al., 2023) have made significant progress. By aligning with advanced LLMs (Touvron et al., 2023; Brown et al., 2020; Liu et al., 2024a) and leveraging high-quality instruction data (Li et al., 2024; Tong et al., 2024) for end-to-end training, LVLMs have greatly expanded their capabilities in tasks such as question answering and reasoning, achieving notable breakthroughs across various domains. Among these advancements, numerous open-source LVLMs have contributed through extensive research in data construction, alignment methods, model architecture, *.etc*. Currently, InternVL-2.5 (Chen et al., 2024a) and Qwen2.5-VL (Bai et al., 2025) stand as the leading LVLM series, gradually closing the gap with closed-source (Achiam et al., 2023) models and even surpassing them on challenging benchmarks like MMMU (Yue et al., 2024). Beyond these achievements, there is a growing focus on more challenging object localization tasks (Bai et al., 2025), such as visual grounding and object detection. While LVLMs have surpassed expert models in simpler fine-grained localization tasks like Referring Expression Comprehension (REC) (Kazemzadeh et al., 2014), they still lag significantly behind specialized models in complex and dense object detection tasks. Although some studies, such as Griffon (Zhan et al., 2024a) and Lumen, (Jiao et al., 2025) have explored this area, they remain limited to supervised fine-tuning,

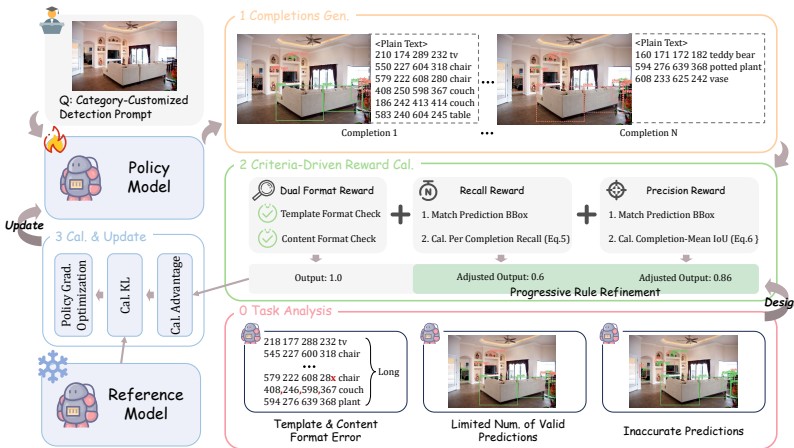

Figure 2: Vision-R1 framework. Green boxes denote correct predictions; red indicates errors. Solid lines represent model output; dashed lines show ground truth.

which offers limited performance gains. As object localization serves as a fundamental capability for enabling more advanced reasoning in LVLMs, it presents both a key research direction and a major challenge. In this paper, we further explore reinforcement learning-based post-training to enhance the performance of state-of-the-art LVLMs on more demanding object localization tasks.

## 2.2 VISION-LANGUAGE REINFORCEMENT LEARNING

With the advancement of LVLMs, researchers have begun exploring reinforcement learning methods to better align these models with human preferences, inspired by the success of reinforcement learning in LLMs (Ouyang et al., 2022; Chen et al., 2024c; Rafailov et al., 2023). The first application in LVLMs named RLHF (Sun et al., 2023), aims to reduce hallucinations by iteratively optimizing model responses based on human feedback. To further enhance alignment and simplify training, Direct Preference Optimization (DPO) (Rafailov et al., 2023) is introduced, allowing models to be trained directly on human-annotated preference data. Since then, various preference optimization algorithms (Yu et al., 2024a;b) have been developed to improve dialogue capabilities, mitigate hallucinations, *.etc*. As LVLMs continue to advance, some methods (Dong et al., 2024; Wang et al., 2024) have also attempted to leverage reinforcement learning to enhance long-sequence reasoning. Despite reducing computational costs compared to pretraining while improving model performance, these approaches still rely on manually annotated preference data (Zhang et al., 2025) and reward model training, making them resource-intensive and challenging. Inspired by the success of the rule-based GRPO (Shao et al., 2024) method in DeepSeek-R1 (Guo et al., 2025), we explore its application in the vision-language domain, where instruction datasets with precise annotations inherently align with human preferences. Our work shows that rule-based reinforcement learning, guided by visual feedback, can significantly enhance object localization tasks, beyond closed-set object detection for specialized models (Pinto et al., 2023), without requiring re-annotated preference data or reward model training. This further highlights its potential for broader applications in LVLMs.

## 3 VISION-R1

In this section, we systematically introduce vision-anchored R1-like reinforcement learning algorithm Vision-R1, a success extension of the GRPO (Shao et al., 2024) reinforcement learning algorithm to the vision field. We start with brief preliminaries about the rule-based GRPO algorithm, which is the source of success R1 models and our foundations. Then, we detail the pivotal component of Vision-R1 algorithm criteria-driven reward function in Section 3.2, specifically the criteria-driven reward function. Moreover, we introduce the progressive rule refinement strategy in Section 3.3. We illustrate the framework of Vision-R1 in Figure 2.

## 3.1 PRELIMINARIES

Building on the success of GRPO in enabling self-evolving, multi-domain reasoning within DeepSeek-R1 (Guo et al., 2025), this reinforcement learning algorithm provides valuable insights to both the language and vision communities. Since its supervision is based solely on the final outcome, GRPO is especially suited for tasks with explicit, objective answers. Unlike other preference optimization methods relying on reward models or value models, it significantly reduces memory overhead for LVLMs. Furthermore, GRPO computes the relative advantages within a group of completions for a given sample, eliminating the need for manually annotated preference data. We further detail its training procedure and optimization loss as follows.

Given an initial model to be optimized, GRPO begins by initializing a trainable policy model $\pi_\theta$ and a frozen reference model $\pi_{ref}$. For a given sample $q$, the old policy model $\pi_{\theta_{old}}$ first generates a group of completions $\{o_1, o_2, ..., o_N\}$. Then, the reward function $f_{reward}$ computes the whole group rewards $\{r_1, r_2, ..., r_N\}$, which are further used to calculate the advantage $A_i$ of each completion with the group by:

$$A_i = \frac{r_i - mean(\{r_j\}_{j=1}^N)}{std(\{r_j\}_{j=1}^N)} \tag{1}$$

After the reference model computes the logits to output each completion given the question, the policy model $\pi_\theta$ is optimized by maximizing the following objective:

$$\mathcal{J}_{GRPO}(\theta) = \frac{1}{N} \sum_{i=1}^N (min(\frac{\pi_\theta(o_i|q)}{\pi_{\theta_{old}}(o_i|q)} A_i, clip(\frac{\pi_\theta(o_i|q)}{\pi_{\theta_{old}}(o_i|q)}, 1-\epsilon, 1+\epsilon)A_i) - \beta \cdot \mathcal{KL}(\pi_\theta(o_i|q)|\pi_{ref}(o_i|q))$$
$$\tag{2}$$

where $N$ is the number of completions in one group and $\beta$ and $\epsilon$ are the hyper-parameters. This objective motivates the model to tend to produce the completion with a higher advantage within a group, but not to stray too far away from the initial model.

## 3.2 CRITERIA-DRIVEN REWARD FUNCTION

Previous approaches (Shao et al., 2024; Guo et al., 2024) have primarily focused on domains such as mathematics and coding, where answers are often summarized using structured templates and evaluated through character-level matching. In contrast, vision-language tasks inherently have definitive answers, and object localization tasks typically do not involve intermediate steps but directly output the final result. While object localization tasks have clear objectives that identify all objects of interest, such visual feedback does not require strict character-level matching. Simply applying previous matching-based reward overlooks the unique characteristics of vision tasks and their feedback, as well as the advantages of reinforcement post-training that operates at the completion level.

To address this, we investigate to design a reward function that accounts for both the nature of object localization tasks and the limitations of current LVLMs in handling them. As shown in the task analysis in Figure 2, LVLMs (Bai et al., 2025; Zhan et al., 2024a; Chen et al., 2024a) face three major challenges in object localization tasks. First, in multi-instance, long-sequence predictions, they often fail to follow instructions correctly, leading to formatting errors. Second, the model produces an insufficient number of valid predictions, failing to detect all mentioned objects. Third, it struggles with small or challenging objects, resulting in inaccurate predictions. Besides the formatting errors, the latter two issues are typically evaluated in object detection. Therefore, we propose a criterion-driven reward function, incorporating dual-format reward, recall reward, and precision reward to comprehensively assess model performance and incentivize improvement.

**Box-prioritized Prediction Matching.** LVLMs outputs object coordinates as textual sequences for object localization tasks due to the unified sequence modeling. To compute rewards based on visual feedback, we first convert these textual sequences into coordinate-based visual feedback as mentioned earlier. Existing LVLMs that support object localization tasks typically follow a fixed sequence representation for object coordinates, such as the plain-text format shown in Figure 2. Based on this representation, we extract individual objects from the sequence. However, object localization tasks often involve multiple objects, requiring exact matches between predictions and ground truth. To address this in training, we unify all object localization tasks under the general framework of object

detection and conduct matching before computing rewards. Unlike detection expert models, LVLMs do not generate class probabilities and are generally less precise in bounding box accuracy, despite correctly predicting object categories. Based on our experiments, we introduce a simplification to the Hungarian matcher (Carion et al., 2020), prioritizing box-based loss for alignment. As indicated in Equation 3, after matching, each predicted instance contains coordinates, a category label, and an Intersection over Union score (IoU).

$$\{P_m^i\}_{m=1}^M = extract\_match(o_i)$$
$$P_m^i = \{[x1, y1, x2, y2]_m^i, label_m^i, IoU_m^i\}$$

(3)

**Dual Format Reward.** Previous methods introduce format rewards to encourage adherence to predefined templates for easy answer extraction. Different from these methods, as illustrated in the first challenge, LVLMs directly output results for object localization tasks, but fall short in long-sequence prediction with both content and template format errors. To address this, we design the dual-format reward. For each completion $o_i$, the template-format checking $f_{tem}$ will verify whether the completion follows the designated template format, such as JSON-format coordinates structure in Qwen2.5-VL (Bai et al., 2025). Once met, we further validate the numerical content to ensure it adheres to coordinate constraints, indicated as $f_{cont}$, such as staying within valid bounds and correctly placing decimal points. We adopt a binary reward scheme, assigning a reward of 1 only when the prediction fully satisfies both format and content criteria as follows:

$$reward_{DF}(o_i) = \begin{cases} 1, & \text{if } f_{tem} = 1 \wedge f_{cont} = 1 \\ 0, & \text{otherwise} \end{cases}$$

(4)

**Recall Reward.** Recall is a crucial metric in object localization tasks, reflecting whether a model can predict all instances of interest as comprehensively as possible without omission. As shown in Figure 2, unlike specialized localization models, LVLMs typically predict fewer confirmed but fewer valid instances than the actual number. Therefore, it is essential to incorporate recall quality into the evaluation of completion to encourage the model to identify all targets as it can. As shown in Equation 5, we follow the definition of recall in object detection and design a recall-based reward for each predicted completion. When the IoU of a matched predicted instance exceeds the predefined threshold $\xi_0$, it is considered a valid prediction. The recall reward is the ratio of valid predictions in all GTs.

$$reward_{recall}(o_i) = \frac{num(Valid\ Predictions)}{num(GT)}$$

(5)

**Precision Reward.** Unlike the global perspective of recall, the precision reward focuses on the quality of the predicted instances of each completion for the third challenge. The precision reward works in conjunction with the recall reward: while the latter encourages the model to predict as many relevant instances as possible, the former ensures that the predictions are as accurate as possible. To directly motivate models to predict high-quality bounding boxes, we define the precision reward as the average IoU of all valid predictions:

$$reward_{prec}(o_i) = \frac{\sum_{m=1}^M [(IoU_m^i \geq \xi_0) \cdot IoU_m^i]}{M}$$

(6)

The overall reward for each completion $o_i$ is the sum of all three rewards to comprehensively assess the completion anchored on the visual task criteria.

$$reward = reward_{DF} + reward_{recall} + reward_{prec}$$

(7)

### 3.3 PROGRESSIVE RULE REFINEMENT STRATEGY

In localization tasks, accurately predicting a bounding box with high IoU to the ground truth is challenging, especially in dense scenes. This difficulty may lead to similar completion rewards for different predictions within the same group, limiting the model's optimization. To address this, we propose a progressive rule refinement strategy, inspired by curriculum learning (Bengio et al., 2009) and human learning processes, which dynamically adjusts reward calculation criteria during training for continuous performance improvement. As shown in Figure 2, this strategy is applied to both

recall and precision rewards, refining their final values for computing the advantage $A_i$. It consists of two key components: differentiation policy and Staged Progression policy.

**Differentiation.** The differentiation strategy focuses on increasing the contrast in the mapping between predictions and actual rewards. Unlike the previous linear mapping, we penalize predictions with low recall and average IoU while granting full rewards to those with relatively high recall and IoU. This adjustment encourages the model to generate high-quality responses within its current capability for optimal rewards. We denote the penalty threshold as $\xi_1$ and the full reward threshold as $\xi_2$, with the differentiation strategy expressed as Eq. 8. We apply this strategy to each instance for the precision reward for better stability, and directly adjust the recall reward for one completion.

$$f(x) = \begin{cases} 1, & \text{if } x \geq \xi_2 \\ 0, & \text{elif } x < \xi_1 \\ x, & \text{otherwise} \end{cases} \tag{8}$$

**Staged Progression.** Providing beginners with an easier-to-achieve standard and gradually increasing the difficulty as their capability improves is a common learning strategy. We incorporate this principle into our design to encourage continuous model improvement and prevent reward hacking. The training process is divided into two phases: initial learning and advanced learning, based on training steps (STEP). In the initial phase, we set relatively low TP thresholds $\xi_0$ and reward criteria $\xi_1, \xi_2$, referring to the threshold settings in object detection evaluations with 0.5, 0.5, and intermediate 0.75. With advancing, we tighten the criteria by adjusting the thresholds to their previous upper bounds: 0.75, 0.75, and 0.9. Since achieving perfectly accurate bounding box predictions is nearly impossible in object localization tasks, we set $\xi_2$ slightly below 1. Through these strategy adjustments, the model can achieve continuous learning and improvement over time.

## 4 EXPERIMENTS

### 4.1 IMPLEMENTATION DETAILS

**Model Setting.** We integrate Vision-R1 with several advanced LVLMs to verify the broad effectiveness of Vision-R1. Specifically, we implement Vision-R1 based on the latest Qwen2.5-VL-7B (Bai et al., 2025) and Griffon-G-7B (Zhan et al., 2024a) models. Qwen2.5-VL-7B is the latest and most comprehensive multimodal large model, demonstrating competitive object localization capabilities in addition to its advanced VQA performance. In contrast, Griffon-G is the first LVLM to approach the performance of specialized localization models. Given their differing localization abilities, we select these two models to evaluate the effectiveness of our method across different model proficiency levels. As a post-training reinforcement learning approach, we directly fine-tune the open-source models using our constructed dataset of 49K samples which we introduce below. Training is conducted with the open-source Open-R1 (Face, 2025) and its multimodal variant framework (Chen et al., 2025), utilizing the default configuration. Specifically, we set $\beta$ to 0.2 and train for 1 epoch with the learning rate of 1e-6. For the comparison method SFT, we use the same data and fine-tune each model for 1 epoch with the learning rate of 2e-6 and batch size of 128. For rapid evaluation, we employ VLMEvalKit (Duan et al., 2024) and Griffon (Zhan et al., 2024b).

**Training Data.** As previously mentioned, Vision-R1 does not require human-annotated preference data and can be directly trained using question-answer pairs with precise answer annotations. To construct the reinforcement learning data, we carefully curate samples from a previously fine-annotated object localization instruction dataset. During the curation process, we adhere to two key principles: diversity and challenge. Ultimately, we construct a 49K reinforcement learning dataset, consisting of 30K object detection samples, 9K visual grounding samples, and 10K Referring Expression Comprehension samples, as object detection is generally more challenging than the other two tasks. Within each data category, we ensure that approximately 50% of the samples are challenging, featuring a greater number of object categories and instances, as well as a proportion of negative samples. A detailed illustration of the dataset is provided in the Appendix.

### 4.2 MAIN RESULTS ON OBJECT LOCALIZATION

**Setup.** We provide extensive experimental results on a wide range of object localization benchmarks, which challenge the model to accurately detect and localize objects across diverse and complex

Table 1: Object localization results on common detection benchmark MSCOCO val2017 (Lin et al., 2014) and ODINW-13 (Li* et al., 2022) benchmarks. We follow ODINW evaluation of (Bai et al., 2025) using the visual grounding setting and report the $Avg.\ mAP$, which indicates the average mAP on all 13 evaluation datasets.

| Type | Model | Res. | MSCOCO Val2017 | | | ODINW-13 |
|---|---|---|---|---|---|---|
| | | | $AP_{50}$ | $AP_{75}$ | $mAP$ | $Avg.\ mAP$ |
| Specialists | Faster RCNN-FPN (Ren et al., 2016) | 1022 | 58.6 | 40.9 | 37.9 | - |
| | DAB-DETR (Liu et al., 2022) | 1333 | 60.3 | 39.8 | 38.0 | - |
| | DETR (Carion et al., 2020) | 1333 | 62.4 | 44.2 | 42.0 | - |
| | Pix2Seq (Chen et al., 2021) | 1333 | 61.0 | 45.6 | 43.0 | - |
| | GroundingDINO (Liu et al., 2024c) | 1333 | - | - | 46.7 | 55.0 |
| Generalist | Griffon-13B (Zhan et al., 2024b) | 448 | 40.6 | 25.1 | 24.8 | - |
| | Griffon v2 (Zhan et al., 2024c) | 1022 | 54.3 | 41.2 | 38.5 | - |
| | Lumen (Jiao et al., 2025) | 448 | 53.2 | 35.8 | 35.3 | - |
| | InternVL2.5-8B (Chen et al., 2024a) | Dynamic | 11.9 | 19.4 | 12.1 | 20.2 |
| | InternVL2.5-78B (Chen et al., 2024a) | Dynamic | - | - | - | 31.7 |
| | Qwen2.5-VL-72B (Bai et al., 2025) | Dynamic | - | - | - | 43.1 |
| | Gemini 1.5 Pro (Team et al., 2024) | - | - | - | - | 36.7 |
| | Griffon-G-7B (Zhan et al., 2024a) | 1022 | 57.4 | 42.8 | 40.2 | 43.8 |
| | + SFT | | 57.4 | 43.3 | 40.5 | 45.3 |
| | + Vision-R1 | | 59.3 | 45.0 | **42.0 (+1.8)** | **46.3 (+2.5)** |
| | Qwen2.5-VL-7B (Bai et al., 2025) | Dynamic | 27.3 | 18.0 | 17.7 | 37.0† |
| | + SFT | | 36.1 | 24.3 | 23.6 | 35.0 |
| | + Vision-R1 | | 40.0 | 27.8 | **26.6 (+8.9)** | **46.0 (+9.0)** |

environments, showcasing its advanced object localization abilities. We incorporate several widely recognized and representative in-domain datasets, spanning dense object detection and real-world scene localization. COCO (Lin et al., 2014) serves as a rigorous and well-acknowledged benchmark for assessing multi-object localization in dense scenes. ODINW-13 (Li* et al., 2022) covers 13 distinct real-world settings with rare object categories, testing the model's capacity to apply its knowledge for object inference in practical scenarios. We also assess methods' generalization ability on out-of-domain untrained localization datasets in challenging scenarios. We employ four Non-overlapping subsets from ODINW (Li* et al., 2022) individually.

**In-domain Object Localization.** The results in Tab. 1 demonstrate the broad effectiveness of the Vision-R1 in object localization tasks. When applied to the Griffon-G model, which excels in object detection, Vision-R1 further improves its performance by 1.8 on COCO and achieves an average mAP increase of 2.5 on ODINW-13. This significantly outperforms the state-of-the-art Qwen2.5-VL-72B on ODINW-13 and brings Griffon-G-7B closer to the performance of specialized vision models. When integrated with the Qwen2.5-VL-7B model, which has relatively weaker localization capabilities, Vision-R1 yields even more substantial improvements, boosting COCO object detection performance by 8.9 points and achieving an 8.7-point gain on ODINW, surpassing the performance of its larger 72B counterpart. Compared to the Supervised Fine-Tuning method, Vision-R1 consistently outperforms it by an average of 1.25 and 7 points on the two models, respectively. Notably, SFT reduces Qwen2.5-VL-7B's performance on ODINW-13, possibly due to over-fitting when training with limited data. These results highlight Vision-R1's strength in enhancing the LVLMs' object localization capabilities with limited training data across different models and scenarios, particularly benefiting weaker models.

**Out-of-domain Object Localization.** As introduced in the setup, we incorporate four non-overlapping datasets from ODINW for out-of-domain localization evaluation. Unlike traditional out-of-domain detection setups, we relax the constraint that both images and object categories must be entirely unseen during training. Given the large-scale training data of LVLMs, strictly ensuring complete novelty is challenging; we here define an experiment setting where either the object category or the scene is absent from the post-training stage to assess generalization ability. As shown in Table 2, Vision-R1 improves performance when integrated with the Griffon-G-7B and Qwen2.5-VL-7B

Table 2: Result on out-of-domain datasets collected from non-overlapping ODINW (Li* et al., 2022), where BB indicates BoggleBoards, MDC indicates MountainDewCommercial, TC indicates ThermalCheetah, and Ve indicates Vector. We follow the grounding setting in (Bai et al., 2025) for evaluation.

| Method | BB | MDC | TC | Ve | Avg. |
|---|---|---|---|---|---|
| GroundingDINO (Liu et al., 2024c) | 0.8 | 18.2 | 12.9 | - | - |
| InternVL2.5-8B (Chen et al., 2024a) | 0.1 | 0.0 | 0.7 | 6.7 | 1.9 |
| Griffon-G-7B (Zhan et al., 2024a) | 3.4 | 28.1 | 8.9 | 8.2 | 12.2 |
| + SFT | 2.3 | 13.7 | 9.4 | 24.0 | 12.4 |
| + Vision-R1 | 3.9 | 41.5 | 7.8 | 24.1 | **19.3 (+7.1)** |
| Qwen2.5-VL-7B (Bai et al., 2025) | 4.5 | 3.8 | 7.8 | 51.3 | 16.9 |
| + SFT | 8.4 | 6.5 | 8.3 | 48.3 | 17.9 |
| + Vision-R1 | 8.2 | 13.7 | 9.9 | 54.8 | **21.7 (+4.8)** |

Table 3: Ablation study on different box matchers for LVLMs.

| Matcher Choice | $mAP$ | $AR100$ |
|---|---|---|
| Box-only | 42.1 | 54.2 |
| Box & Label | 41.9 | 53.4 |

Table 4: Ablation study on Reward Function Design.

| P | R | $mAP$ | $AP^{50}$ | $AP^{75}$ | $AR100$ |
|---|---|---|---|---|---|
| Baseline | | 40.2 | 57.4 | 42.8 | 52.2 |
| ✓ | | 41.5 | 55.6 | 45.1 | 49.6 |
| ✓ | ✓ | 42.1 | 58.7 | 45.3 | 54.2 |

models, achieving average gains of 7.1 and 4.8, respectively. Notably, it surpasses expert models on BoggleBoards and MountainDewCommercial, further demonstrating its strong generalization capability beyond specific datasets. While the SFT performs competitively in challenging scenarios involving heatmaps, *.etc*, where LVLMs initially struggle, it exhibits a significant performance drop in more common scenes compared to the base model. This suggests that SFT lacks robust generalization, whereas Vision-R1 effectively enhances both in-domain and out-of-domain performance.

## 4.3 ABLATION STUDIES

In this section, we provide comprehensive experiments to validate the design of Vision-R1, underscoring our key contributions. Unless otherwise specified, we conduct ablation experiments using the detection data from our constructed dataset, which can be regarded as a general form of localization tasks, making the experiments more representative and broadly applicable.

**Discussion on Different Matcher Approaches.** As mentioned in Section 3.2, prior box matching is typically based on Hungarian matching, which minimizes loss by considering both box accuracy and category prediction scores. However, unlike detection expert models, LVLMs do not rely on a predefined category set with probabilistic outputs, and directly produce deterministic category labels instead. Building on this, we simplify the assignment process by either considering only box accuracy or incorporating both box accuracy and category correctness. As shown in Table 3, the two approaches exhibit a limited significant performance difference, with the box-only matching method performing slightly better. We attribute this to the strong classification ability of LVLMs, which rarely misclassify objects when predicting a small number of targets. Matching solely based on bounding boxes helps the model recall more objects, leading to a slight performance gain after training by enabling more accurate predictions.

**Effectiveness of Reward Function Design.** To comprehensively evaluate the design of our reward function, we first conduct an ablation study to compare the effects of the three reward components. Among them, the dual format reward primarily serves as feedback for some completions where the model fails to follow the expected format or content template. Therefore, we focus our ablation comparison on precision reward and recall reward. When excluding the recall reward, we introduce a binary prediction count reward, which grants a reward only when the predicted number of instances matches the ground truth. This prevents the model from continuously generating redundant outputs. As shown in Table 4, when only precision is considered, the model produces higher-quality bounding

boxes, leading to an increase in all levels of AP. However, the number of recalled instances decreases. With the introduction of the recall reward, the model's recall rate increases by 2% compared to the baseline, and the overall mAP further improves by 0.6, demonstrating that our design to integrate recall and precision leads to more effective performance.

**Effectiveness of Progressive Rule Refinement.** The progressive rule refinement strategy serves as a mechanism to encourage continuous model improvement. In our experiments, we set and fixed $\xi$ following object detection evaluation criteria while adjusting STEP to determine the optimal transition point for the advanced phase. To examine the impact of different configurations, we conducted a comparative study on the Griffon-G-7B model, evaluating three settings where STEP was set to 1/3, 1/2, and 1, and tested

Table 5: Ablation study on Progressive Rule Refinement. STEP indicates the training stage where refinement begins.

| STEP | $mAP$ | $AP^{50}$ | $AP^{75}$ | $AR100$ |
|---|---|---|---|---|
| Baseline | 40.2 | 57.4 | 42.8 | 52.2 |
| 1/3 | 41.5 | 58.0 | 44.8 | 54.6 |
| 1/2 | 42.1 | 58.7 | 45.3 | 54.2 |
| 1 | 39.9 | 57.0 | 42.6 | 56.7 |

the performance on COCO. As shown in Table 5, adjusting the model at STEP = 1/2 yielded the best performance, whereas keeping STEP = 1 (*i.e.*, no adjustment) resulted in performance lower than the baseline. Our analysis suggests that for the Griffon-G model, which initially possesses strong localization abilities, recall has a greater impact during training. As a result, it achieved an $AR^{100}$ of 56.7. However, without progressive reward adjustments, the model generated a large number of lower-quality bounding boxes, contributing to more false positives in AP metrics, ultimately reducing $mAP$ slightly below the baseline. When adding our strategy, it will suppress these low-quality boxes chasing after more high-quality boxes. While for the comparably weak Qwen2.5-VL-7B model, the situation is different with STEP = 1 yielding the best results as reported, which we detail in the Appendix. These overall results validate the importance and effectiveness of our progressive rule refinement strategy, demonstrating that properly tuning the training process leads to meaningful performance improvements.

**Effects on General QAs.** Vision-R1 aligns LVLMs with subjective annotations that human naturally prefers to advance their object localization capabilities. However, it is also well preferred to remain LVLMs' strong general QA capabilities. We evaluate both LVLMs integrated with Vision-R1 in Table 1 and 2 across various general VQAs, including knowledge (AI2D (Kembhavi et al., 2016)), commonsense (GQA (Hudson & Manning, 2019)), chart (ChartQA

Table 6: Ablation on generalization QA capabilities with results reproduced by VLMEvalKit under the same setting.

| Method | GQA | AI2D | ChartQA | SEED |
|---|---|---|---|---|
| Griffon-G-7B | 64.6 | 70.1 | 68.7 | 71.7 |
| + SFT | 63.5 | 70.5 | 67.5 | 72.2 |
| + Vision-R1 | 64.8 | 70.3 | 68.8 | 71.8 |

(Masry et al., 2022)), and interdisciplinary (SEED (Li et al., 2023b)) domains. As shown in Table 6, training with Vision-R1 results in minimal fluctuations in general QA performance, maintaining a performance similar to the baseline model, while SFT methods show a significant drop. This indicates that our method significantly enhances object localization without heavily compromising general QA abilities. Moreover, the improvement in object localization leads to a performance boost on object-perception-based commonsense tasks like GQA, further showcasing the advantages of our approach. We also provide experimental results for Qwen2.5-VL-7B in the appendix, further demonstrating the effectiveness of our method.

## 5 CONCLUSION

In this paper, we introduce Vision-R1, a novel reinforcement learning algorithm for LVLMs that combines a vision criterion-driven reward function and a progressive rule refinement strategy to enhance their object localization capabilities. By designing this algorithm, we present a human-annotating-free approach to leverage abundant instruction data with subjective and definite responses embodied to boost LVLMs' localization performance. Comprehensive evaluation across various benchmarks under diverse scenarios demonstrates the generalized effectiveness of our method, encouraging more research to equip LVLMs with advanced, precise object localization capabilities to support complex tasks and real-life applications. We provide more discussion on the limitations and broader impact in the appendix.

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
