In this supplementary material, we provide details on dataset construction, task templates, evaluation data, and detailed results. We also present a further analysis of the progressive adjustment strategy.

## A  TRAINING DATA CONSTRUCTION

Table 7: Details of constructed training dataset.

| Type | Num. | Source |
|---|---|---|
| Object Detection | 30K | COCO (Lin et al., 2014) |
| Visual Grounding | 9K | ODINW (Li* et al., 2022), V3Det (Wang et al., 2023) |
| REC | 10K | RefCOCO (Kazemzadeh et al., 2014), Visual Genome (Krishna et al., 2017) |

Our method does not rely on human preference data but instead selects localization-related instruction data for training. As described in Section 4.1, we curate data from open-source localization instruction datasets, primarily covering object detection, visual grounding, and REC. The data sources and their quantities are summarized in Table 7, with the selection process detailed below:

**Object Detection** For object detection, we select data from MS COCO (Lin et al., 2014), which includes a diverse range of scenes and object categories. We define images with more than 10 instances as hard cases, considering the category count and the current capabilities of LVLMs. We follow the distribution of raw data, and sample one-third of the data from COCO's difficult and easy samples. Since object detection serves as a general representation of localization tasks, it is inherently more complex and can help improve localization performance across other tasks. As a result, we prioritize object detection data, selecting a total of 30K samples.

**Visual Grounding** To enhance the model's ability to localize objects across diverse scenes and categories, we incorporate ODINW and V3Det datasets, which cover over 13K categories. We convert these datasets into a visual grounding format, ensuring consistency with our task requirements. Additionally, following Griffon-G, we include a small portion of multi-category annotations from V3Det. From these two datasets, we collect 5K and 4K samples each.

**Referring Expression Comprehension** For REC tasks, we start with the widely used RefCOCO dataset (Kazemzadeh et al., 2014) and further consider more general cases where a referring expression corresponds to multiple objects, as highlighted in GRefCOCO (He et al., 2023). We collect 5K samples from RefCOCO and additionally extract multi-object referring expressions from the Visual Genome dataset (Kembhavi et al., 2016).

This dataset selection process ensures a balanced and comprehensive training foundation for improving LVLM performance on challenging object localization tasks.

## B  TRAINING DETAILS

We use 2 nodes of NVIDIA H100 GPUs for the training. The total training time is about 26 hours for the Griffon-G-9B model and 16 hours for Qwen2.5-VL-7B model.

## C  TASK TEMPLATES

After constructing the dataset, we further build the instruction templates for object localization tasks. Our approach primarily follows the task templates used during the instruction fine-tuning phase of the selected models (Zhan et al., 2024a; Bai et al., 2025). For tasks not explicitly covered in the instruction fine-tuning phase, we adapt and adjust templates based on related task formats. Finally, we create five instruction templates for model training and evaluation, as summarized in Table 8.

## D  DETAILED RESULTS

As demonstrated in Section 4.2, we follow the Qwen2.5-VL to evaluate our method on ODINW-13 (Li* et al., 2022), in which the evaluation sets follow the setting of the mmdetection toolkit (Chen

Table 8: Training and evaluation templates for each model and task.

| Model | Task | Template |
|---|---|---|
| Griffon-G | Object Detection | Examine the image for any objects from the category set. Report the coordinates of each detected object. The category set includes {category list}. |
| | Visual Grounding | Locate the exact position of {category} in the picture, if you can. |
| | REC | Can you point out {ref expression} in the image and provide the coordinates of its location? |
| Qwen2.5-VL | Object Detection | Locate every item from the category list in the image and output the coordinates in JSON format. The category set includes {category list}. |
| | Visual Grounding REC | Locate every {category} in the image and output the coordinates in JSON format. |

Table 9: Detailed results on ODINW-13 dataset.

| Model | ODINW-13 | | | | | | | | | | | | | |
|---|---|---|---|---|---|---|---|---|---|---|---|---|---|---|
| | *AerialDrone* | *Aquarium* | *Rabbits* | *EgoHands* | *Mushrooms* | *Packages* | *PascalVOC* | *Pistols* | *Pothole* | *Raccoon* | *Shellfish* | *Thermal* | *Vehicles* | *AVG* |
| InternVL2.5-8B(Chen et al., 2024a) | 0 | 6.9 | 38.5 | 0.2 | 26.7 | 16.4 | 37.0 | 29.2 | 1.1 | 46.6 | 28.5 | 3.8 | 27.1 | 20.2 |
| Qwen2.5-VL-3B(Bai et al., 2025) | 6.2 | 16.4 | 75.0 | 24.6 | 8.3 | 66.6 | 52.0 | 42.3 | 10.2 | 47.7 | 36.7 | 40.7 | 57.1 | 37.2 |
| Griffon-G-7B (Zhan et al., 2024a) | 9.5 | 18.7 | 74.3 | 25.6 | 39.7 | 56.5 | 59.3 | 55.1 | 11.7 | 51.3 | 51.8 | 49.2 | 67.2 | 43.8 |
| Griffon-G-7B-SFT | 12.9 | 20.7 | 75.5 | 50.6 | 39.9 | 54.6 | 61.2 | 52.8 | 10.9 | 49.5 | 48.1 | 49.5 | 62.9 | 45.3 |
| Griffon-G-7B-Vision-R1 | 12.2 | 21.5 | 75.2 | 48.0 | 50.4 | 54.1 | 62.2 | 56.3 | 17.2 | 48.2 | 40.7 | 55.9 | 59.9 | 46.3 |
| Qwen2.5-VL-7B(Bai et al., 2025) | 7.8 | 20.3 | 73.5 | 32.2 | 7.0 | 57.6 | 49.8 | 48.5 | 7.4 | 40.1 | 42.7 | 38.0 | 56.3 | 37.0 |
| Qwen2.5-VL-7B-SFT | 0.9 | 8.8 | 78.8 | 17.4 | 8.0 | 48.4 | 51.5 | 49.6 | 9.1 | 44.9 | 36.0 | 46.9 | 55.1 | 35.0 |
| Qwen2.5-VL-7B-Vision-R1 | 7.5 | 22.6 | 77.1 | 49.9 | 67.7 | 50.2 | 54.4 | 55.2 | 13.3 | 43.9 | 49.9 | 50.6 | 56.5 | 46.0 |

et al., 2019). Due to the page length limitation, we do not provide the detailed results of all 13 datasets in the main body. To clearly demonstrate Vision-R1's advancement, we list the results of all 13 datasets from different models in Table 9. As seen in the table, Vision-R1 improves these two models on most of the sets by a large margin, which highlights the effectiveness of our method.

# E    ADDITIONAL ABLATION STUDIES

## E.1    FURTHER ANALYSIS OF PROGRESSIVE RULE REFINEMENT STRATEGY

The progressive rule refinement strategy is an effective approach to preventing reward hacking, ensuring continuous performance improvement. As demonstrated in Section 4.3, for the Griffon-G-7B model, which already exhibits relatively strong performance, the model consistently earns high precision rewards on average, while recall is relatively lower, making the recall reward more influential. As recall increases without an improvement in box quality, overall precision remains unchanged or slightly decreases. After employing our strategy, we assign full rewards (value = 1) to high-quality bounding boxes, encouraging the model to refine box accuracy. This, combined with the increased recall, leads to a mAP improvement. By adjusting STEP, we control when the model transitions into the advanced training phase to get the optimal performance.

For Qwen2.5-VL-7B, which has weaker localization capabilities, the optimal STEP setting differs, as shown in Table 10. Given our $\xi$ hyperparameter settings, Qwen2.5-VL struggles to consistently achieve an average precision and recall above 0.75. In this case, setting STEP = 1 (no adjustment) yields the best performance. When adjusting the reward criteria midway, the model fails to meet the stricter standards, leading to reduced training efficiency and weaker performance. However, all settings still outperform the baseline, demonstrating the effectiveness of our method.

In summary, the STEP hyperparameter makes our progressive rule refinement strategy highly adaptable to models with different capability levels. For stronger models, progressive rule adjustment during training continuously reinforces optimization. For weaker models, a later STEP setting allows the model to first meet the initial reward criteria before transitioning. If the model fails to meet the predefined adjustment threshold within the given data volume, setting STEP = 1 is a viable option. By following these guidelines, users can tune hyperparameters based on specific models and datasets to achieve optimal performance.

Table 10: Experiments on Progressive Rule Refinement with Qwen2.5-VL-7B.

| STEP | $mAP$ | $AP^{50}$ | $AP^{75}$ | $AR100$ |
|------|-------|-----------|-----------|---------|
| Baseline | 17.7 | 27.3 | 18.8 | 29.8 |
| 1/2 | 23.3 | 32.0 | 25.2 | 28.9 |
| 1 | 26.6 | 40.0 | 27.8 | 36.7 |

### E.2 GENERAL VQAS RESULTS FOR QWEN2.5-VL MODEL

We provide ablation results on generalization QA capabilities for Qwen2.5-VL-7B here in Table 11. The result with a slight difference, further demonstrating the effectiveness of our method, that our method significantly enhances object localization without heavily compromising general QA abilities.

Table 11: Ablation on generalization QA capabilities. Results are all produced by VLMEvalKit under the same setting.

| Method | GQA | AI2D | ChartQA | SEED |
|--------|-----|------|---------|------|
| Qwen2.5-VL-7B | 58.8 | 80.8 | 87.5 | 76.7 |
| + SFT | 53.5 | 80.2 | 83.4 | 76.2 |
| + Vision-R1 | 61.0 | 80.3 | 86.0 | 75.5 |

## F QUALITATIVE ANALYSIS

We provide a quantitative analysis to better demonstrate the effectiveness of our approach. For this analysis, we use the Qwen2.5-VL-7B model, which shows a more significant performance improvement, making the qualitative impact of our method more evident.

As shown in Figure 3, the original Qwen2.5-VL-7B model often produces redundant outputs, suffers from a high number of missed detections, and generates imprecise bounding boxes in detection tasks. After training with our method, the model eliminates redundant and invalid outputs, significantly improves recall, and maximizes the retrieval of relevant targets, ultimately achieving more precise localization.

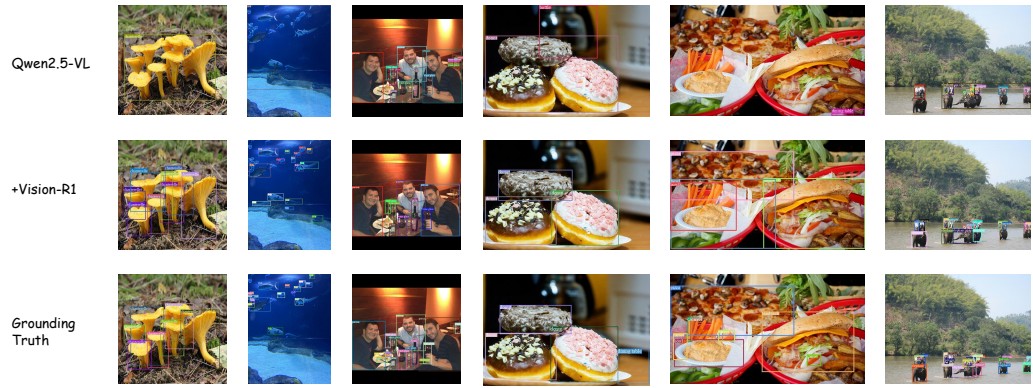

Figure 3: Qualitative analysis results using Qwen2.5-VL-7B

## G LLM USAGE

Our use of a Large Language Model (LLM) in this work was limited to providing word-level hints and occasional suggestions for sentence improvement. This is in accordance with the ICLR's Code of Conduct on LLM usage.