# OpenReview forum: "Vision-R1: Evolving Human-Free Alignment in Large Vision-Language Models via Vision-Guided Reinforcement Learning"
_ICLR.cc/2026/Conference — ICLR 2026 Conference Withdrawn Submission_

### Official Review · Reviewer_d2HS · 2025-10-31

**Soundness:** 2
**Presentation:** 1
**Contribution:** 2
**Rating:** 2
**Confidence:** 4

**Summary:**

This paper proposes Vision-R1, a vision-guided reinforcement learning framework for large vision-language models (LVLMs). The authors aim to remove the need for human-annotated preference data and train LVLMs using vision-anchored, rule-based rewards. The key components include (1) a criterion-driven reward function combining dual-format, recall, and precision rewards derived from visual task logic, and (2) a progressive rule refinement strategy that dynamically tightens reward thresholds to avoid reward hacking and encourage continuous improvement. Experiments on object localization (COCO, ODINW-13) show that Vision-R1 improves mAP over SFT.

**Strengths:**

Improvements are solid. Vision-R1 yields gains on both in-domain (COCO, ODINW-13) and out-of-domain subsets, demonstrating generalization beyond supervised fine-tuning.

Broad evaluation. The paper evaluates across multiple models (Qwen2.5-VL-7B, Griffon-G-7B) and tasks, and includes many ablation studies.

**Weaknesses:**

**Limited originality.** Similar ideas (e.g., Visual-RFT, VLM-R1, Perception-R1) appeared months earlier and also use rule-based or verifiable rewards for LVLMs. The paper does not clearly differentiate its contribution or cite these related works. The claimed novelty of “vision-guided RL” thus feels incremental.

**Over-claimed scope.** Although the title and abstract emphasize “vision-guided reinforcement learning,” the experiments are limited to a single task type—object grounding/detection. There is no evidence the method generalizes to broader vision tasks such as captioning, classification, or vision reasoning, which weakens the claimed generality.

**Poor reward generalization.** The so-called visual feedback is essentially a mixture of IoU, recall, and precision, which are intrinsic metrics of grounding tasks. Once leaving detection-based settings, these signals become meaningless, limiting transferability to other vision domains.

**Misattributed contribution.** The claim of “eliminating the need for specialized reward models and handcrafted preference datasets” stems directly from the GRPO framework itself, not from innovations in this paper. This should be clarified as inherited advantage rather than contribution.

**Weak novelty in Box-prioritized Matching.** The proposed “Box-prioritized Prediction Matching” is trivial since modern LVLMs (e.g., Qwen2.5-VL) already output bounding boxes natively. The matching strategy adds little theoretical or practical innovation.

**Ambiguous data source.** The “49K reinforcement learning dataset” is described as curated but without specifying its provenance, licensing, or whether it mixes public benchmarks—this undermines reproducibility and potential fairness evaluation.

**Methodological clarity issues.** Several equations (e.g., Eq. 3–8) lack variable definitions and intuitive explanations, making it difficult to follow how rewards are computed and normalized.

**Questions:**

1.Data origin and fairness: Where exactly do the 49K “curated” samples come from? Are they extracted from COCO/ODINW or newly annotated? How do you ensure no data leakage into evaluation sets?

2.Comparison baselines: Why are strong post-training baselines like Visual-RFT, VLM-R1, or Perception-R1 not included for comparison? They seem highly relevant.

3.Clarify the genuine novelty. Many similar works (e.g., Visual-RFT, VLM-R1, Perception-R1) have already explored R1-like or GRPO-style reinforcement learning for LVLMs. Could the authors explicitly articulate what is new in Vision-R1 beyond applying GRPO to visual grounding?

4.The remaining questions can refer to the weaknesses.

---

### Official Review · Reviewer_FEcG · 2025-11-01

**Soundness:** 3
**Presentation:** 3
**Contribution:** 3
**Rating:** 4
**Confidence:** 4

**Summary:**

This paper proposes a criterion-driven reward function and progressive rule refinement strategy tailored for localization tasks to enhance LVLM training on visual grounding.

**Strengths:**

1. The writing is clear, and the experiments are solid.
2. The paper addresses a problem in LVLMs, enhancing object localization capabilities without requiring expensive preference annotation.
3. The evaluation includes multiple benchmarks (COCO, ODINW-13) and both in-domain and out-of-domain testing.
4. The method is only evaluated on object localization tasks.

**Weaknesses:**

1. The results in Table 6 are unstable compared to SFT.
2. The reward supervision for visual perception is biased and may suffer from reward hacking, as there is no ground truth exists.
3. The task is too narrow, using LVLMs for tasks like object localization.
4. The IoU thresholds (0.5, 0.75, 0.9) appear to be chosen based on "threshold settings in object detection evaluations", but no systematic analysis justifies why these specific values are optimal for RL training in LVLMs.

**Questions:**

What is the impact of this method on LVLMs on other general VQA benchmarks? Intuitively, if an LVLM's fine-grained visual localization capability is substantially enhanced, shouldn't this improve performance on other visual instruction-following tasks as well? What's the value of a general 7B LVLM over a specialized location model?

---

### Official Review · Reviewer_wAjV · 2025-11-01

**Soundness:** 3
**Presentation:** 3
**Contribution:** 2
**Rating:** 2
**Confidence:** 4

**Summary:**

Inspired by DeepSeek-R1, this paper introduces Vision-R1, a vision-guided reinforcement learning algorithm
for Large Vision-Language Models (LVLMs). The method leverages curated instruction data, eliminating the
need for human-annotated preference data and specialized reward models. It introduces a criterion-driven
reward function that integrates dual-format, recall, and precision rewards, along with a progressive rule
refinement strategy to dynamically adjust reward criteria during training. Experiments on in-domain and out-
of-domain object localization benchmarks demonstrate its consistent performance improvements.

**Strengths:**

- The reward function is carefully designed to address the core challenges of LVLMs in object localization
tasks (such as visual grounding and object detection), which include formatting errors, low recall, and
inaccurate localization. The combination of dual-format, recall, and precision rewards effectively guides
the model's optimization direction.
- On multiple challenging benchmarks (e.g., MSCOCO, ODINW-13), this method achieves significant and
consistent performance improvements on both in-domain and out-of-domain data, especially for models
with weaker localization capabilities (e.g., Qwen2.5-VL-7B), proving its effectiveness and generalization
ability.
- While focusing on enhancing object localization capabilities, this method does not harm the model's
performance on general visual question answering (e.g., GQA, ChartQA), and even shows slight
improvement on some tasks, indicating a balanced optimization.
- The progressive rule refinement strategy (including differentiation and staged progression) draws
inspiration from curriculum learning, effectively addresses the problem of sparse reward signals in object
detection tasks, promotes continuous model improvement, and mitigates reward hacking.

**Weaknesses:**

- Experiments are conducted only on 7B parameter models. Whether this method is equally effective and
scalable for larger-scale LVLMs (e.g., 72B) that are continuously emerging remains unclear.
- While the main experiments show that Reinforcement Learning (RL) outperforms Supervised Fine-Tuning
(SFT), the paper lacks a deep analysis of the underlying reasons. What specific optimization behaviors or
internal representational changes does RL enable that are unattainable with SFT? A more convincing
mechanistic explanation is needed, going beyond merely presenting performance numbers.
Furthermore, what leads to the differing impacts of Vision-R1 on Griffon-G-7B versus Qwen2.5-VL-7B?
- The paper does not clearly state whether there is any overlap (in images or categories) between its
constructed 49K training dataset and the various test sets (especially subsets drawn from large datasets,
like ODINW), which raises concerns about potential data leakage.
- Key thresholds in the reward function (such as the IoU threshold ξ₀, and ξ₁, ξ₂ in the progressive strategy)
are primarily referenced from traditional evaluation standards in the object detection field, but lack
ablation studies or hyperparameter searches within the reinforcement learning framework to prove their
optimality.
- Besides general visual question answering, the paper does not evaluate whether the model degrades on
other important capabilities (such as mathematical reasoning, code generation, complex multi-turn
dialogue). It remains unclear whether the reinforcement of localization capability comes at the cost of
other reasoning abilities.
- Unclear details:
    - The meaning of the "Res." column in Table 1 is not explicitly explained (speculated to be image
resolution), but the huge differences in resolution across models and the impact on fair comparison
are not discussed.
    -  The statement of whether all compared methods (SFT baselines, other models) were trained (or re-
trained) on the exact same 49K dataset could be more precise.
    - The paper omits critical details regarding the concrete training setup for Vision-R1. It is unclear
whether parameter-efficient fine-tuning methods (e.g., LoRA) or full-parameter fine-tuning were
employed. Furthermore, which specific components or modules of the model were optimized during
reinforcement learning? These implementation details are essential for both reproducibility and a
thorough understanding of the work.

The biggest concern of this work regarding the core motivation and effectiveness of the approach: Although the paper
addresses drawbacks of traditional RLHF, a key objective is to enhance LVLMs' visual capabilities to match
or surpass specialist models on specific tasks while maintaining generality. However, the experimental
results do not fully support this objective:
    - On core tasks like object detection, the model's performance remains significantly behind specialist
models (e.g., DETR) even after proposed optimization.
    - On some VQA tasks, the improvements brought by RL are limited and do not consistently surpass
SFT.
    - Given the existence of mature specialist models for object localization, and given the currently
demonstrated limited advantage in general capabilities, what is the necessary unique advantage of
employing LVLMs with the proposed RL scheme? The authors need to more compellingly argue the
indispensable value of their paradigm compared to simply "using a specialist model" or "performing
SFT for a specific task."

**Questions:**

- Given that experiments are limited to 7B models, what are the authors' insights or preliminary results
regarding the scalability of Vision-R1 to much larger LVLMs (e.g., 20B+ parameters)? Are there any
potential obstacles?
- Beyond the performance metrics, can the authors provide a more mechanistic explanation for why RL
outperforms SFT? For instance, does the reward signal lead to qualitatively different attention patterns or
internal representations? Also, what factors might explain the differing efficacy of Vision-R1 between
Griffon and Qwen2.5-VL?
- Could the authors please detail the data curation process to explicitly confirm that there is no overlap
between the 49K training dataset and all test sets (especially ODINW subsets), thereby alleviating data
leakage concerns?
- Was any ablation study or hyperparameter search conducted to validate the chosen reward thresholds
(ξ₀, ξ₁, ξ₂)? How sensitive are the final results to these specific values?
- To better assess the broader impact of localization-focused RL tuning, can the authors evaluate model
performance on other key capabilities such as mathematical reasoning (e.g., on MathVista) or coding
(e.g., on MMCode)?

Other feedback related to the weaknesses.

---

### Official Review · Reviewer_X2am · 2025-11-04

**Soundness:** 3
**Presentation:** 3
**Contribution:** 3
**Rating:** 4
**Confidence:** 5

**Summary:**

This paper presents Vision-R1, a post-training alignment framework for large vision-language models (LVLMs) that aims to enhance visual grounding and localization capabilities without relying on human preference data or reward models. Instead, the authors design a criterion-driven reward composed of three interpretable components—dual-format correctness, recall, and precision (mean IoU)—directly aligned with object detection objectives. To further stabilize training and prevent reward hacking, a progressive rule refinement strategy gradually tightens evaluation thresholds, forming a curriculum-like schedule. The method is trained via GRPO using around 49K curated localization samples for a single epoch. Experiments on COCO and ODINW-13 show consistent improvements over both base and SFT models (e.g., +8.9 AP on COCO for Qwen2.5-VL-7B) while maintaining general QA performance across benchmarks such as GQA, AI2D, and ChartQA. Ablation studies confirm the necessity of the rule refinement process and each reward component.

**Strengths:**

The paper targets post-training alignment for LVLMs without human preference data or learned reward models, proposing Vision-R1 with a criterion-driven reward and progressive rule refinement—an appealingly simple, principled alternative to preference/RM pipelines. The criterion-driven reward decomposes into dual-format (template + numeric), recall, and precision (mean IoU) terms, aligning directly with object-localization objectives rather than string-match proxies—this is an excellent fit for LVLM localization outputs. The progressive rule refinement strategy (differentiation + staged progression with tighter IoU/TP thresholds) is a neat curriculum-style device to sharpen gradients, discourage reward hacking, and continue improving once basic format compliance is achieved.

**Weaknesses:**

1. The main experiments are conducted only on two LVLMs (Griffon-G-7B and Qwen2.5-VL-7B). While both are representative, this narrow coverage makes it difficult to assess the generality and robustness of the proposed training framework. Demonstrating consistent improvements on more architectures would better support claims of broad applicability.

2. The paper’s title “Vision-R1” implies a general post-training alignment framework for vision-language models. However, the method and experiments focus almost exclusively on visual grounding / object localization. Although the authors show that general QA performance does not degrade, the work does not actually enhance general vision understanding beyond grounding. A more precise title such as “Vision-Grounding-R1” would more accurately reflect the paper’s actual scope and avoid misleading readers’ expectations.

3. The proposed training pipeline is implemented solely with GRPO. It remains unclear whether Vision-R1’s improvements stem from the framework itself or from GRPO-specific advantages. The authors should provide evidence that the framework is optimizer-agnostic, for example by reproducing results with a different RL algorithm or by including an ablation across optimizers.

4. OOD generalization is a critical issue for visual grounding tasks. Although the paper briefly reports ODINW results, the OOD definition is relatively loose and the discussion is limited. A more systematic analysis—clearly distinguishing in-domain vs. out-of-domain splits, and quantifying generalization gaps—would greatly strengthen the empirical evidence and highlight the claimed robustness benefits.

**Questions:**

See weaknesses above.

---

### Official Review · Reviewer_z8Dd · 2025-11-06

**Soundness:** 3
**Presentation:** 3
**Contribution:** 2
**Rating:** 2
**Confidence:** 3

**Summary:**

The paper presents Vision-R1, a reinforcement learning framework aimed at improving Large Vision-Language Models (LVLMs) on vision-based tasks without relying on human-annotated preferences or learned reward models. Vision-R1 employs vision-guided, rule-based rewards that combine dual-format, recall, and precision components into a single performance metric, along with a progressive rule refinement strategy that gradually increases task difficulty. Applied to object-localization tasks, the method fine-tunes Griffon-G-7B and Qwen2.5-VL-7B on a 49K-sample dataset, achieving substantial performance gains and better generalization across domains. Moreover, Vision-R1 maintains or slightly improves general VQA accuracy, offering a cost-effective and robust alternative to human-in-the-loop reinforcement learning.

**Strengths:**

The methodology is carefully described. The multi-part reward function is justified by the task analysis, and the progressive refinement is a sensible curriculum strategy.

**Weaknesses:**

Although the paper presents an interesting extension of reinforcement learning ideas to vision-language tasks, addressing the following concerns could further strengthen the work:

1.  **Baselines:** It would be beneficial to compare Vision-R1 with additional relevant methods such as Vision-RFT [1], Ground-R1 [2], Rex-Thinker [3], and GRIT [4] to provide a more comprehensive evaluation.


2. **Cold Start Analysis:** As highlighted in the DeepSeek-R1 paper, the cold start phase is critical for the GRPO algorithm. An analysis exploring whether this step is equally crucial for vision-language models would offer valuable insights.


3. **Reward Sensitivity (Ablation Study):** An ablation study examining the sensitivity and contribution of each reward component would help clarify the impact of combining dual-format, recall, and precision rewards.


4. **Missing limitations:** Although the authors mention providing further discussion on limitations and broader impacts in the appendix, this section appears to be absent in the paper (I couldn't find it in the Supplementary Material) and should be included for completeness.


5. **Dataset Details:** The paper lacks sufficient details about the curated dataset, including its sources, preprocessing steps, and characteristics, which are important for reproducibility.


6. **Hyperparameter Tuning and Reward Shaping:** Since hyperparameter tuning is crucial for optimization, a comprehensive tuning process and an in-depth analysis of reward shaping during training are recommended. Including relevant plots would further strengthen the empirical evaluation.

---

**References**

[1] Vision-RFT: https://arxiv.org/abs/2503.01785

[2] Ground-R1: https://arxiv.org/abs/2505.20272v1

[3] Rex-Thinker: https://arxiv.org/abs/2506.04034v1

[4] GRIT: https://arxiv.org/abs/2505.15879

**Questions:**

Please read the weaknesses.

---

### Note · Authors · 2025-11-12

I have read and agree with the venue's withdrawal policy on behalf of myself and my co-authors.